# Exosome-Mediated Activation of Neuronal Cells Triggered by γ-Aminobutyric Acid (GABA)

**DOI:** 10.3390/nu13082544

**Published:** 2021-07-25

**Authors:** Ryo Inotsuka, Miyako Udono, Atsushi Yamatsu, Mujo Kim, Yoshinori Katakura

**Affiliations:** 1Graduate School of Bioresources and Bioenvironmental Sciences, Kyushu University, Fukuoka 819-0395, Japan; inotsuka.ryo.840@s.kyushu-u.ac.jp; 2Faculty of Agriculture, Kyushu University, Fukuoka 819-0395, Japan; mudono@grt.kyushu-u.ac.jp; 3International GABA Research Center, Kyoto 615-8245, Japan; a-yamatsu@pharmafoods.co.jp; 4Pharma Foods International Co., Ltd., Kyoto 615-8245, Japan; mujokim@pharmafoods.co.jp

**Keywords:** GABA, exosome, gut-brain interaction, Caco-2, SH-SY5Y

## Abstract

γ-Aminobutyric acid (GABA) is a potent bioactive amino acid, and several studies have shown that oral administration of GABA induces relaxation, improves sleep, and reduces psychological stress and fatigue. In a recent study, we reported that exosomes derived from GABA-treated intestinal cells serve as signal transducers that mediate brain–gut interactions. Therefore, the purpose of this study was to verify the functionality of GABA-derived exosomes and to examine the possibility of improving memory function following GABA administration. The results showed that exosomes derived from GABA-treated intestinal cells (Caco-2) activated neuronal cells (SH-SY5Y) by regulating genes related to neuronal cell functions. Furthermore, we found that exosomes derived from the serum of GABA-treated mice also activated SH-SY5Y cells, indicating that exosomes, which are capable of activating neuronal cells, circulate in the blood of mice orally administered GABA. Finally, we performed a microarray analysis of mRNA isolated from the hippocampus of mice that were orally administered GABA. The results revealed changes in the expression of genes related to brain function. Gene Set Enrichment Analysis (GSEA) showed that oral administration of GABA affected the expression of genes related to memory function in the hippocampus.

## 1. Introduction

γ-Aminobutyric acid (GABA) is a naturally occurring nonprotein amino acid that is one of the principal inhibitory neurotransmitters of the central nervous system (CNS) [1]. The human GABA_B_ receptor-a member of the class C family of G-protein-coupled receptors (GPCRs)-mediates inhibitory neurotransmission [2]. It has been reported that GABAergic synapses constitute 20–50% of all synapses present in the CNS [3] and play important roles in information encoding and behavioral control, regulation of motor learning, and motor functions [4,5]. Generally, genes characterized by GABAergic function are down-regulated during normal aging in humans [6], and in certain studies involving animals, it has been suggested that GABA function declines with age. For example, the number of GABA-immunoreactive neurons declines with age in the hippocampus, inferior colliculus, and striate visual cortex of animals [7,8,9]. There are also reported reductions in GABA levels, GABA release, and GABA receptor binding from the baseline levels with aging [8]. When the concentration of GABA in the brain diminishes below a certain threshold, various neurological disorders including epilepsy, seizures, convulsions, and Alzheimer disease have also been reported [10,11,12]. Furthermore, it has been recently reported that serum GABA levels are decreased in various diseases including stroke [13]. These studies suggest that GABA has health-promoting effects that may also be related to the enhancement of brain function.

The development of functional foods containing GABA has been actively studied. In a recent study, a randomized, double-blind, placebo-controlled clinical trial involving healthy Japanese adults was reported, where administration of 100 mg of GABA resulted in increased participant memory and spatial cognitive function [14]. Indeed, several studies have shown that oral administration of GABA induces relaxation, improves sleep, and reduces psychological stress and fatigue [15,16,17]. Additionally, healthy adults who consumed 50 mg of GABA dissolved in a beverage reported less occupational fatigue after completion of required tasks [18]. However, in view of the lack of evidence regarding the blood–brain barrier permeability of GABA, the mechanisms through which GABA might exert these beneficial effects in humans remain unclear. It is considered that the oral intake of GABA exerts such effects through an indirect pathway [19].

Furthermore, GABA has been reported to have a variety of health promoting effects including immunomodulatory, anti-diabetes, anti-cancer, anti-oxidant, and so on [20,21,22,23]. Future research is expected to focus not only on neurological and psychological disorders linked to a decrease in amount of GABA, but also on the systemic health-promoting effects of GABA and its molecular basis.

In our previous study, we clarified the molecular basis of GABA-induced gut–brain interactions and reported that exosomes derived from GABA-treated intestinal cells serve as signal transducers that mediate brain–gut interactions [24]. Exosomes are a family of particles released from the cell that are delimited by a lipid bilayer, and attention has recently been focused on the role of exosomes as biomarker candidates for diagnosis, prognosis and even therapeutic tools of various diseases [25]. In the present study, we clarified that exosomes derived from serum of mice administered GABA as well as from GABA-treated Caco-2 cells activated neuronal cells, and that GABA administration changes the expression of memory-related genes in hippocampus. 

## 2. Materials and Methods

### 2.1. Cell Culture and Reagents

The human colorectal cancer cell line Caco-2 (ATCC, Manassas, VA, USA) and the human neuroblastoma cell line SH-SY5Y (ATCC) were cultured in Dulbecco’s modified Eagle’s medium (DMEM; Nissui, Tokyo, Japan) supplemented with 10% heat-inactivated fetal bovine serum (FBS, Life Technologies, Gaithersburg, MD, USA) at 37 °C in an atmosphere containing 5% CO_2_. γ-Aminobutyric acid (GABA) was purchased from Abcam (Cambridge, UK). 5-Aminoimidazole-4-carboxamide 1-β-d-ribofuranoside (AICAR) and retinoic acid (RA) were purchased from FUJIFILM Wako Pure Chemical Corp. (Osaka, Japan).

### 2.2. Exosome Isolation and Treatment

Firstly, Caco-2 cells (1.4 × 10^5^ cells/mL) were cultured in DMEM containing 10% Exosome-depleted FBS media supplement heat inactivated (System Bioscience, Mountain View, CA, USA) and 500 or 1000 μM GABA. After 24 h of culture, the MagCapture Exosome Isolation Kit PS (FUJIFILM Wako Pure Chemical Corp.) was used to isolate exosomes from the media of Caco-2 cells, according to the manufacturer’s instructions. Exosomes were isolated from mouse serum using ExoQuick Exosome Precipitation Solution (System Biosciences, Palo Alto, CA, USA), according to the manufacturer’s instructions. SH-SY5Y cells (2.0 × 10^5^ cell/mL) were cultured for 24 h, and treated with exosomes (equivalent to 90 ng protein) derived from GABA-treated Caco-2 cells for 24 h.

### 2.3. Quantitative Evaluation of Neurite Growth

SH-SY5Y cells were seeded onto a μClear fluorescence black plate (Greiner-Bio One, Tokyo, Japan), fixed with 4% paraformaldehyde for 15 min, and blocked with blocking buffer (1 × PBS, 5% goat serum, and 0.3% Triton X-100) for 1 h. The cells were subsequently incubated with Milli-Mark Pan Neuronal Marker (Merck Millipore, Billerica, MA, USA) at 25 °C overnight. After washing with PBS, the cells were stained with Alexa Fluor 555 goat anti-rabbit IgG antibody (Thermo Fisher Scientific, Inc., Waltham, MA, USA) for 1 h at 25 °C. After washing with PBS, cells were further stained with Hoechst 33342 (Dojindo, Kumamoto, Japan) for 15 min, and neurite length was measured using the IN Cell Analyzer 2200 (GE Healthcare, Amersham Place, UK), as previously described [24]. The total neurite length for each cell is shown in the figure.

### 2.4. Mitochondria

Cells were stained with 250 nM MitoTracker Red CMXRos (Thermo Fischer Scientific) at 37 °C for 30 min, and subsequently with 200 nM MitoTracker Green FM (Thermo Fischer Scientific) at 37 °C for 30 min. Finally, the cells were stained with Hoechst 33342 at 37 °C for 30 min. Stained cells were analyzed using IN Cell Analyzer 2200 (GE Healthcare, Amersham Place, UK) to quantitatively determine the number, area, and activity of mitochondria using IN Cell Investigator high-content image analysis software (GE Healthcare). 

### 2.5. Quantitative Reverse Transcriptase-Polymerase Chain Reaction (RT-qPCR)

RNA was prepared from cells using the High Pure RNA Isolation kit (Roche Diagnostics GmbH, Mannheim, Germany) according to the manufacturer’s protocols. RT-qPCR was performed using the GoTaq 1-Step RT-PCR System (Promega, WI, USA) and Thermal Cycler Dice Real Time System TP-800 (Takara). Samples were analyzed in triplicate. The PCR primer sequences used were as follows: human β-actin (*ACTB*) forward primer 5′-TGGCACCCAGCACAATGAA-3′ and reverse primer 5′-CTAAGTCATAGTCCGCCTAGAAGCA-3′: human brain-derived growth factor (*BDNF*) forward primer 5′-GTCAAGTTGGGAGCCTGAAATAGTG-3′ and reverse primer 5′-AGGATGCTGGTCCAAGTGGTG-3′: peroxisome proliferator-activated receptor γ coactivator 1-*α* (*PGC-1α*) forward primer 5′-GCTGACAGATGGAGACGTGA-3′ and reverse primer 5′-TAGCTGAGTGTTGGCTGGTG-3′: human *NESTIN* forward primer 5′-ACTGGGAAGGAGGAGGTGGT-3′ and reverse primer 5′-CACACTGGCTCCCTCAACCA-3′: human neurofilament medium chain (*NEFM*) forward primer 5′-AGACATCCACCGGCTCAAGG-3′ and reverse primer 5′-CGACGCCTCCTCGATGTCT-3′. β-actin was used as a housekeeping gene. Samples were normalized and analyzed by the ΔΔCT method [26].

### 2.6. miRNA Microarray Assay

The expression profile of miRNA in exosomes prepared using the MagCapure Exosome Isolation Kit PS was evaluated using microarray analysis with an Affimetrix GeneChip miRNA 4.0 Array (Affymetrix, Santa Clara, CA, USA). Total RNA was isolated from exosomes using TRIzol Reagent (Thermo Fisher Scientific) and purified using the miRNeasy Mini Kit (Qiagen, Valencia, CA, USA). Subsequent operations were outsourced to Cell Innovator (Fukuoka, Japan), a commercial contract analysis provider. To identify up- or down-regulated genes, we calculated ratios (non-log scaled fold-change) from the normalized intensities of each gene for comparisons between control and experimental samples. Then, we established criteria for regulated genes: (up-regulated genes) ratio ≥ 1.3-fold; (down-regulated genes) ratio ≤ 0.77 [27]. miRNA target genes were predicted using miRWalk (http://mirwalk.umm.uni-heidelberg.de, access on 20 February 2021). To determine significantly over-represented gene ontology (GO) categories and significantly enriched pathways, we used tools and data provided by the Database for Annotation, Visualization, and Integrated Discovery (DAVID, http://david.abcc.ncifcrf.gov, access on 20 February 2021) [28,29].

### 2.7. mRNA Microarray Assay

The mRNA expression profile was evaluated using a DNA microarray (SurePrint G3 Human Gene Expression 8 × 60 K v.3, Agilent). Total RNA was isolated from SH-SY5Y cells and mouse brains using Isogen II (Nippon Gene, Tokyo, Japan). The subsequent operations were outsourced to Cell Innovator. We then established criteria for significantly up- or down-regulated genes: up-regulated genes, Z-score ≥ 2.0 and ratio ≥ 1.5-fold; down-regulated genes: Z-score ≤ −2.0 and ratio ≤ 0.66-fold. Significantly over-represented GO categories and enriched pathways were determined as described above. Gene set enrichment analysis (GSEA) was performed to determine the enrichment score (ES), which indicates the degree to which each gene set is overrepresented at the top or bottom of a ranked list of genes.

### 2.8. Integrated Analysis

We then performed integrated analysis of miRNAs with altered expression in Exo-GABA and of mRNAs with altered expression in SH-SY5Y cells in response to Exo-GABA treatment, and selected genes.

### 2.9. Animal Experiments

Six-week-old female C57BL/6 mice were obtained from KBT Oriental Co. Ltd. (Saga, Japan) and allowed to adapt for 2 weeks, with food and water provided ad libitum. All mouse experiments and protocols were in accordance with the Guide for the Care and Use of Laboratory Animals and were approved by the Ethics Committee on Animal Experimentation (Kyushu University; approval number: A28-187-0). One group was composed of five mice. GABA solution was orally administered to mice once a day in 100 µL (200 mg/kg) doses using a sonde. The control group received 100 µL of sterile water orally. After 7 days of treatment, cardiac blood samples were collected from the mice, and serum samples were prepared. Serum was used for exosome isolation by using the MagCapture Exosome Isolation Kit PS. Simultaneously, hippocampal tissue was collected from mouse brains.

### 2.10. Statistical Analysis

All experiments were performed at least three times, and the corresponding data are shown. The results are presented as means ± standard deviation. Multiple comparisons between groups were carried out by one-way ANOVA with Tukey’s post-hoc test. Statistical significance was defined as *p* < 0.05 (* *p* < 0.05; ** *p* < 0.01; *** *p* < 0.001).

## 3. Results

### 3.1. Exosomes Derived from GABA-Treated Caco-2 Cells Activate SH-SY5Y Cells

Caco-2 cells are used in this study because they are known to exhibit differentiation functions such as digestion and absorption. First, we tested whether exosomes derived from GABA-treated Caco-2 cells activated SH-SY5Y cells. Exo-ctrl shows the exosomes derived from non-treated Caco-2 cells. As shown in Figure 1A, although the treatment of Caco-2 cells with 500 or 1000 µM GABA did not change the amount of exosomes that could be isolated (data not shown), exosomes derived from GABA-treated Caco-2 cells (Exo-GABA) exhibited elongated neurites in SH-SY5Y cells as compared to Exo-ctrl. However, the neurite outgrowth effect of Exo-GABA was lower than that of the positive control retinoic acid (RA). Furthermore, Exo-GABA was found to augment the expression of *PGC-1α*, a master gene of mitochondrial biogenesis, and increased the number, area, and activity of mitochondria in SH-SY5Y cells (Figure 1B–E), as compared to Exo-ctrl. 5-Aminoimidazole-4-carboxamide 1-β-d-ribofuranoside (AICAR) is known to induce mitochondrial biogenesis [30], and then used as a positive control. As can be seen from some of the results, exosomes derived from Caco-2 cells treated with 1000 μM GABA were found to be more effective.

Next, we tested the effects of Exo-GABA on the expression of marker genes (brain-derived growth factor, *BDNF*; Nestin; and neurofilament medium chain, *NEFM*) of neuronal cells in SH-SY5Y cells. The results showed that Exo-GABA augmented the expression of these marker genes (Figure 2A–C). These results indicated that Exo-GABA activated SH-SY5Y cells.

### 3.2. Molecular Basis for the Exo-GABA-Induced Activation of SH-SY5Y

We then tried to identify miRNAs contained in Exo-GABA and their target genes that activate SH-SY5Y cells using integrated analysis. Exo-ctrl, Exo-GABA_500_, and Exo-GABA_1000_ show the exosomes derived from non-treated, 500 µM GABA-treated and 1000 µM GABA-treated Caco-2 cells, respectively. First, microarray analysis of miRNAs in Exo-GABA was used to search for miRNAs in exosomes that varied with GABA treatment. The results showed that GABA treatment decreased the expression levels of the four miRNAs in Caco-2 cells (Table 1). The target gene of these four miRNAs can be found in the Appendix A. KEGG pathway analysis showed that all four miRNAs were involved in neuronal cell regulation and activation (Table 1).

Changes in mRNA expression in Exo-GABA-treated SH-SY5Y cells were examined using microarray analysis (Figure 3), and we identified 641 mRNAs whose expression was commonly altered in SH-SY5Y treated with Exo-GABA_500_ and Exo-GABA_1000_. These genes can be directly or indirectly regulated by miRNAs and include both up-regulated and down-regulated genes. Similarly, using DAVID, KEGG pathway analysis estimated the involvement of pathways implicated in neuronal activity (Table 2).

We then performed integrated analysis of miRNAs with altered expression in Exo-GABA_500_ and Exo-GABA_1000_ and of mRNAs with altered expression in SH-SY5Y cells in response to Exo-GABA_500_ and Exo-GABA_1000_ treatment. After comparison of miRNA target genes whose expression was altered both in Exo-GABA_500_ and Exo-GABA_1000_ and those whose expression was altered in SH-SY5Y cells in response to Exo-GABA_500_ and Exo-GABA_1000_, 185 common genes were found, 12 of which were involved in the regulation of brain function (Table 3).

### 3.3. Effects of Exosomes Derived from the Serum of GABA-Treated Mice in SH-SY5Y Cells

First, we tested whether serum-derived exosomes from mice orally administered GABA (seExo-GABA) could induce neurite outgrowth in SH-SY5Y cells. After isolating exosomes from serum of mice administered GABA, SH-SY5Y cells were similarly treated with seExo-GABA (equivalent to 90 ng protein) for 24 h, as mentioned above. The results clearly showed that seExo-GABA induced neurite outgrowth in SH-SY5Y cells (Figure 4A). Furthermore, seExo-GABA augmented the expression of *PGC-1α* and increased the number and area of mitochondria in SH-SY5Y cells, but not activity (Figure 4B–E).

Next, we tested the effects of seExo-GABA on the expression of neuronal cell marker genes in SH-SY5Y cells. The results showed that seExo-GABA augmented the expression of some of marker genes except Nestin (Figure 5A–C). These results indicated that seExo-GABA activated SH-SY5Y cells, and revealed that exosomes, which are capable of activating neuronal cells, circulate in the blood of mice orally administered GABA.

### 3.4. Microarray Analysis of Hippocampal Tissue of Mice Orally Administered GABA

Here, we performed microarray analysis of hippocampal mRNA of mice orally administered GABA. Results showed that in the hippocampus of mice orally administered GABA, 1127 and 249 genes were significantly up- and down-regulated, respectively. Among these genes, many genes were observed to be related to brain function (Table 4 and Table 5). Furthermore, GSEA analysis of these genes showed that oral administration of GABA up-regulated the expression of genes related to memory function in the hippocampus (Figure 6).

## 4. Discussion

Exosomes derived from various types of cells have been reported to induce neuroprotection and neural recovery by modulating the expression of genes, proteins, and miRs in target cells and tissues [31,32]. Emerging evidence shows that exosome-mediated multiple communication axes between various organs such as the brain, heart, kidneys, and intestines, as well as systemic immune responses, can influence health status. In the present study, we showed that GABA activated intestinal cells to secrete exosomes that activate neurons, which may shed light on the exosome-mediated activation of gut–brain interactions caused by GABA. Since it has been reported that GABA_A_ receptor is expressed in Caco-2 cells [33], it will be necessary to verify whether GABA activates Caco-2 cells via its receptors to change the content of exosomes, or whether GABA itself is taken up by cells and incorporated into exosomes to function in the target tissue. Recent research has shown that GABA, a type of amino acid, is found in many fermented foods such as yogurt and pickled vegetables. We have previously shown that carnosine, as well as GABA, activates gut–brain interactions via exosomes, but this is a rare example of exosome-mediated activation of gut–brain interactions by a food component [24,34]. In this study, we clarified that serum exosomes of mice administered GABA (seExo-GABA) as well as exosomes derived from GABA-treated Caco-2 cells (Exo-GABA) activated neuronal cells to induce neurite growth and mitochondrial activation. These results suggest that these exosomes function as mediators that carry signals to activate neurons, and that certain foods including carnosine and GABA can produce these neuron-activating exosomes. This study could lead to the creation of a new research area: the development of foods that control brain function through the secretion of functional exosomes from the gut.

Microarray analysis of miRNA in Exo-GABA and of mRNA in SH-SY5Y cells treated with Exo-GABA identified signaling pathways which might be activated by Exo-GABA and be involved in GABA-induced phenotypes. Integrated analysis of miRNAs with altered expression in Exo-GABA and of mRNAs with altered expression in SH-SY5Y cells in response to Exo-GABA treatment identified 12 genes which were involved in the GABA-induced activation of brain function. In the future study, we would like to clarify the functionality of these 12 genes in the regulation of brain function.

There are many possible mechanisms by which exosomes regulate neuronal function. In a previous study, we reported that carnosine augmented the expression of miR-6769-5p in exosomes derived from carnosine-treated Caco-2 cells, which led to repression of target gene (ATXN1) expression, thereby activating neurons [34]. With further analysis, we would like to clarify the molecular mechanism of GABA-mediated activation of neuronal cells through exosomes. In particular, it is necessary to verify whether exosomes contain GABA or other bioactive parts in the future.

The present study also showed that serum exosomes of mice administered GABA activated neuronal cells (Figure 7). This result indicated that exosomes, which can activate neurons, circulate in the blood of mice after GABA administration. Indeed, exosomes derived from mesenchymal stromal cells may be useful for remodeling and functional recovery of neurons after stroke by transferring miR-133b to astrocytes and neurons and regulating gene expression [35]. Recently, various functions of milk exosomes derived from bovine milk have been reported. The milk exosome has been shown to carry specific miRNAs (miR-148a) to various target cells and to exert various functions on them. A characteristic example of milk exosome functionalities include effects against α-synuclein pathology in Parkinson’s disease and Type 2 diabetes mellitus [36]. In other words, both milk exosomes and GABA-derived exosomes are similar in the sense that they are carried in the bloodstream to the target tissue. These reports indicate that serum exosomes are involved in the regulation of brain function, and suggest that by analyzing serum exosomes after ingestion, the functionality of foodstuffs can be verified. Furthermore, it will be possible to develop diagnostic methods for brain function based on serum exosomes.

To verify the effect of GABA administration on brain function, the effects of GABA on sleep, relaxation, and psychological and physical fatigue have been investigated in humans [15,18,37]. These results indicate that GABA strongly affects the early stages of sleep, reduces both psychological and physical fatigue, and acts as a natural relaxant inducer. These results also suggest that GABA may also contribute to improved brain function, especially memory function. GSEA analysis of genes in the hippocampus of GABA-treated mice revealed that GABA administration could have some effect on brain function through the regulation of genes related to memory function. The detailed molecular mechanisms by which GABA regulate gene expression in the hippocampus need to be clarified in the future.

## 5. Conclusions

In this study, we clarified that serum exosomes of mice administered GABA (seExo-GABA) as well as exosomes derived from GABA-treated Caco-2 cells (Exo-GABA) activated neuronal cells to induce neurite growth and mitochondrial activation. These results suggest that these exosomes function as mediators that carry signals to activate neurons, and that certain foods including GABA can produce these neuron-activating exosomes. This study could lead to the creation of a new research area: the development of foods that control brain function through the secretion of functional exosomes from the gut.

## Figures and Tables

**Figure 1 nutrients-13-02544-f001:**
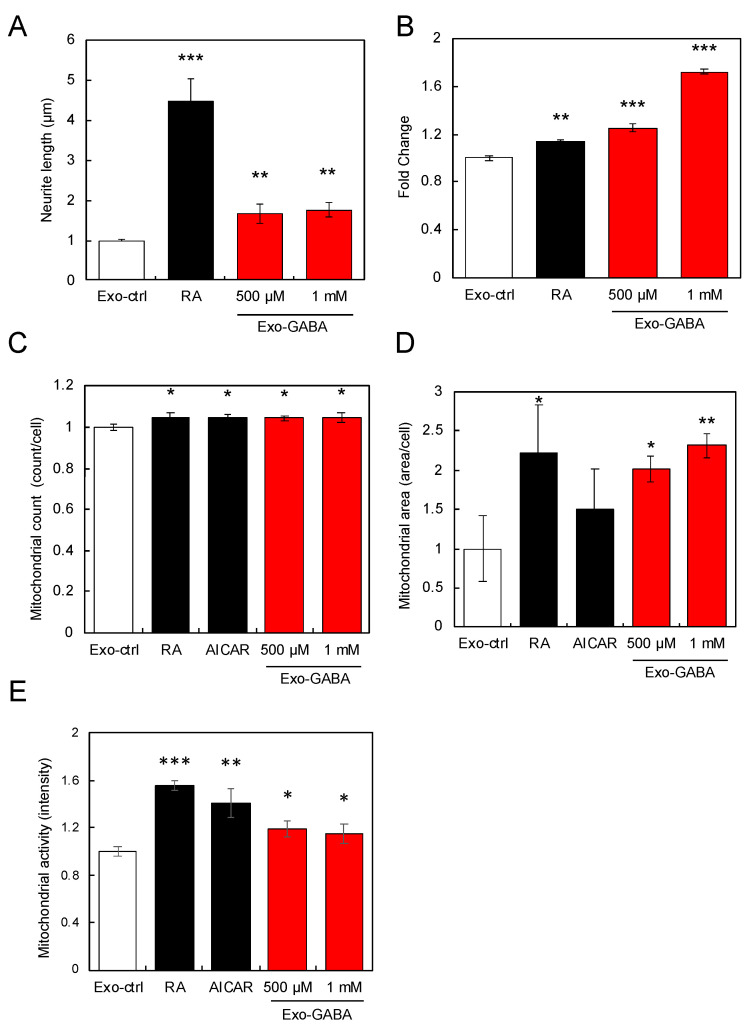
Exosomes derived from GABA-treated Caco-2 cells activate SH-SY5Y cells. Neurite length (**A**) and *PGC-1α* expression (**B**) in SH-SY5Y cells treated with exosomes (equivalent to 90 ng protein) derived from GABA-treated Caco-2 cells. Exo-ctrl shows the exosomes derived from non-treated Caco-2 cells. Count (**C**), area (**D**), and activity (**E**) of mitochondria in SH-ST5Y cells treated with exosomes derived from GABA-treated Caco-2 cells. Retinoic acid (RA) was used as positive control. Multiple comparisons between groups were carried out by one-way ANOVA with Tukey’s post-hoc test. Statistical difference was evaluated by comparing to Exo-ctrl. Statistical significance was defined as *p* < 0.05 (* *p* < 0.05; ** *p* < 0.01; *** *p* < 0.001) (value means ± SEM, *n* = 3).

**Figure 2 nutrients-13-02544-f002:**
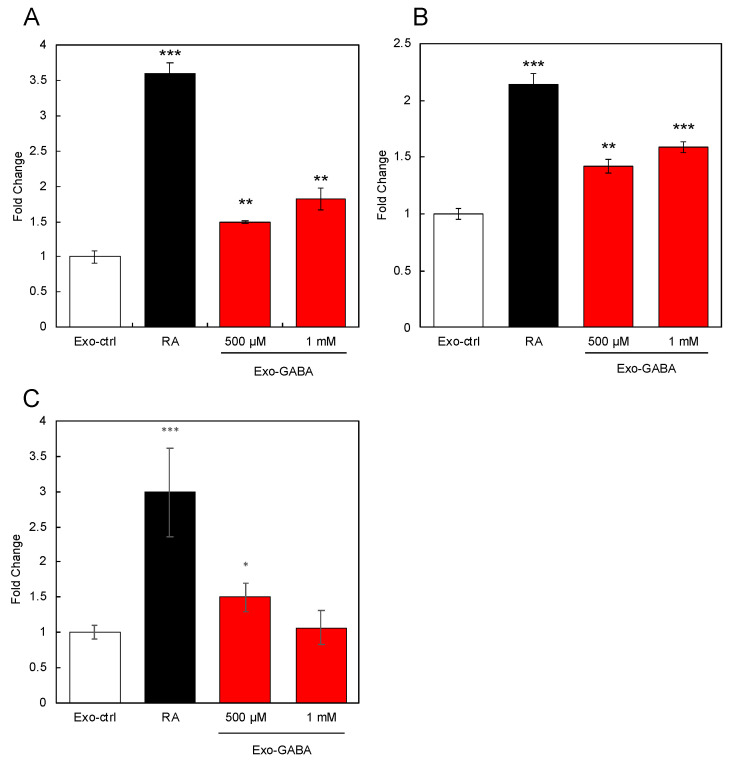
Effects of exosomes derived from GABA-treated Caco-2 cells on gene expression. Expression of brain-derived growth factor (*BDNF*) (**A**), Nestin (**B**), and neurofilament medium chain (*NEFM*) (**C**) in SH-SY5Y cells treated with exosomes derived from GABA-treated Caco-2 cells was evaluated by RT-qPCR. Retinoic acid (RA) was used as positive control. Multiple comparisons between groups were carried out by one-way ANOVA with Tukey’s post-hoc test. Statistical difference was evaluated by comparing to Exo-ctrl. Statistical significance was defined as *p* < 0.05 (* *p* < 0.05; ** *p* < 0.01; *** *p* < 0.001) (value means ± SEM, *n* = 3).

**Figure 3 nutrients-13-02544-f003:**
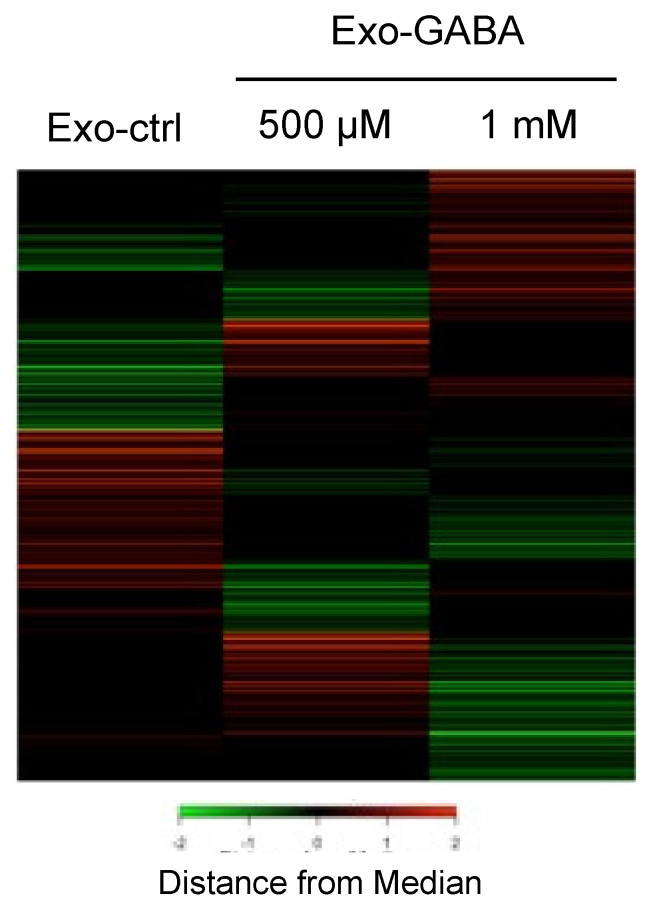
Microarray analysis of genes in exosome-treated SH-SY5Y cells. Heatmap was used to visualize differential expressed genes.

**Figure 4 nutrients-13-02544-f004:**
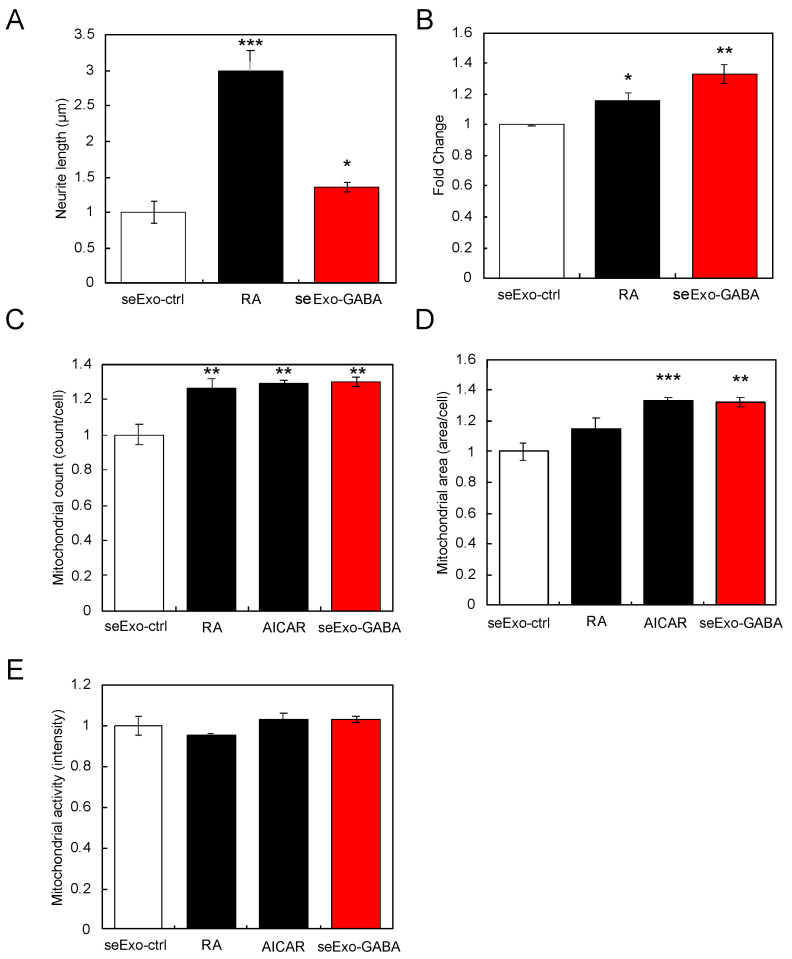
Effects of exosomes derived from serum of GABA-treated mice (seExo-GABA) on SH-SY5Y cells. Exosomes derived from serum of non-treated mice (seExo-ctrl) was used as control. Neurite length (**A**) and *PGC-1α* expression (**B**) in SH-SY5Y cells treated with seExo-GABA. Count (**C**), area (**D**), and activity (**E**) of mitochondria in SH-ST5Y cells treated with seExo-GABA. RA was used as positive control. AICAR is used as positive control for mitochondrial biogenesis. Multiple comparisons between groups were carried out by one-way ANOVA with Tukey’s post-hoc test. Statistical difference was evaluated by comparing to seExo-ctrl. Statistical significance was defined as *p* < 0.05 (* *p* < 0.05; ** *p* < 0.01; *** *p* < 0.001) (value means ± SEM, *n* = 3).

**Figure 5 nutrients-13-02544-f005:**
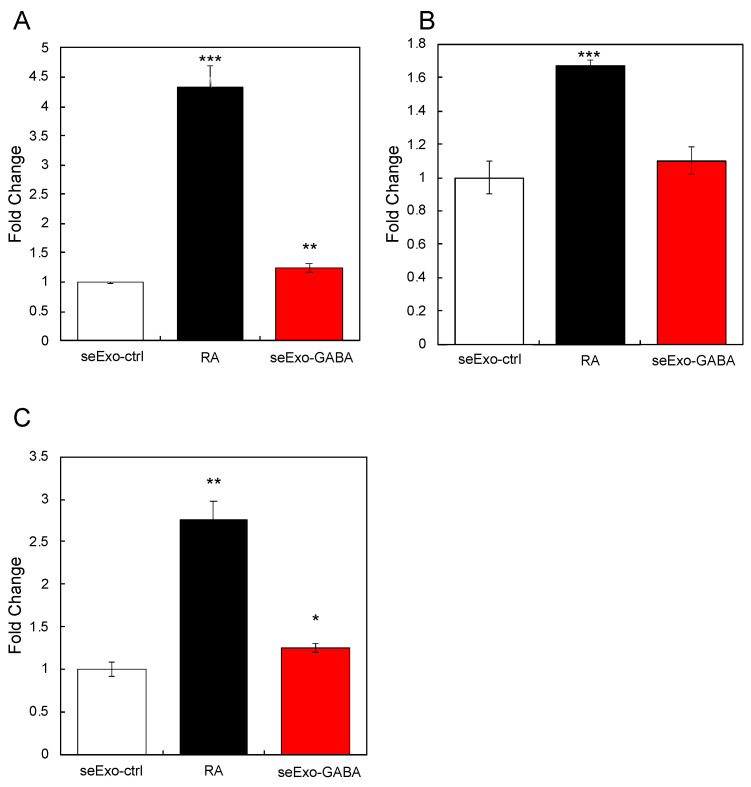
Effects of exosomes derived from serum of GABA-treated mice (seExo-GABA) on gene expression. Exosomes derived from serum of non-treated mice (seExo-ctrl) was used as control. Expression of *BDNF* (**A**), Nestin (**B**), and *NEFM* (**C**) in SH-SY5Y cells treated with seExo-GABA was evaluated by RT-qPCR. RA was used as positive control. Multiple comparisons between groups were carried out by one-way ANOVA with Tukey’s post-hoc test. Statistical difference was evaluated by comparing to seExo-ctrl. Statistical significance was defined as *p* < 0.05 (* *p* < 0.05; ** *p* < 0.01; *** *p* < 0.001) (value means ± SEM, *n* = 3).

**Figure 6 nutrients-13-02544-f006:**
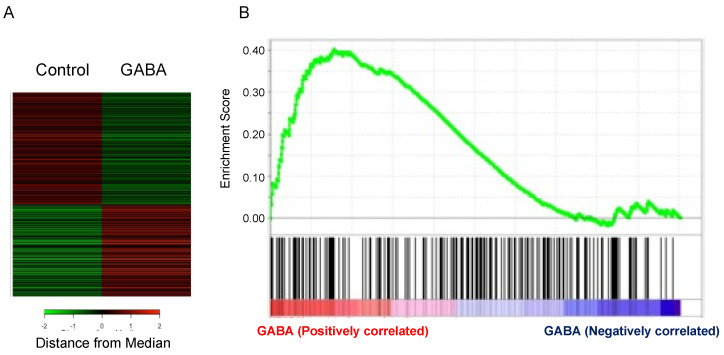
Microarray analysis of genes in the hippocampus of GABA-treated mice. (**A**) Heatmap was used to visualize differential expressed genes. (**B**) Gene set enrichment analysis (GSEA) analysis of gene sets relating to memory in the hippocampus of GABA-treated mice. Enrichment plots (green curve) show the running sum of enrichment score (ES) for memory-related gene set. The score at the peak of the plot is the ES for the gene set. The black bars show where the member of the gene set appear in the ranked list of genes. Each black bar represents a memory-related gene. A predominance of black bars to the left or right side indicates that these genes are up-regulated or down-regulated in the hippocampus of GABA-treated mice.

**Figure 7 nutrients-13-02544-f007:**
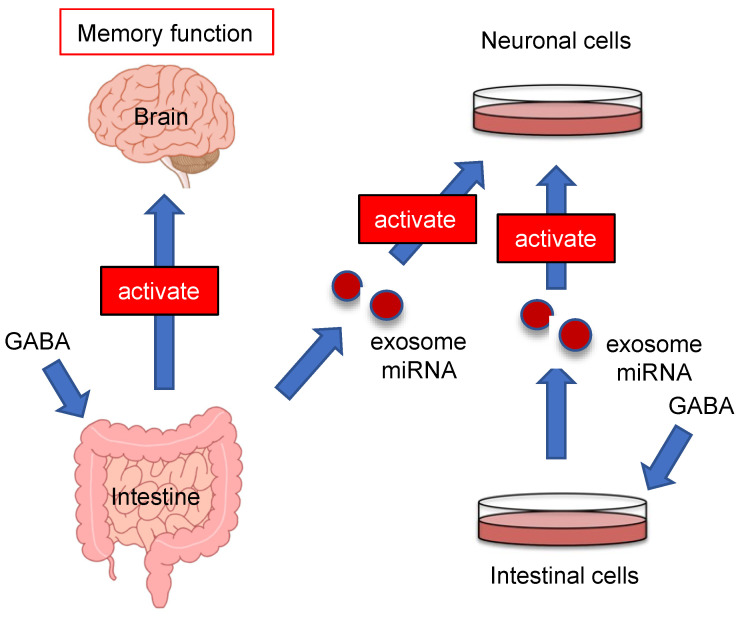
Schematic diagram of GABA function.

**Table 1 nutrients-13-02544-t001:** Functional classification of miRNAs differentially expressed in GABA-treated Caco-2 cells.

miRNA	KEGG Pathway
miR-6732-5p	Calcium signaling pathway, Axon guidance, Neurotrophin signaling pathway
miR-8075	Calcium signaling pathway, Regulation of actin cytoskeleton, Long-term potentiation, Neurotrophin signaling pathway, Axon guidance
miR-3665	Axon guidance, Long-term potentiation, Calcium signaling pathway, Neurotrophin signaling pathway
miR-5787	Axon guidance, Long-term potentiation, Calcium signaling pathway, Neurotrophin signaling pathway, Long-term depression

**Table 2 nutrients-13-02544-t002:** Functional classification of genes differentially expressed in SH-SY5Y cells treated with exosomes derived from GABA-treated Caco-2 cells.

KEGG Pathway	*p*-Value
Cytokine–cytokine receptor interaction	0.0037
Neuroactive ligand–receptor interaction	0.025
Inflammatory bowel disease	0.029

**Table 3 nutrients-13-02544-t003:** Neuronal gene and its function selected by integration analysis.

Gene	Function
CIT	Development of the central nervous system
SLC6A17	Neurotransmitter uptake
MPPED2	Brain development
NPAS3	Neurogenesis
SHANK2	Scaffolding at the synapse
NEUROG1	Neuronal differentiation
RIT1	Neuron development and regeneration
SLC5A7	Depression
GCSAM	Signaling pathway
KCNN2	Regulation of neuronal excitability
KCNK13	Neurotransmitter release
KAT6B	Brain development

**Table 4 nutrients-13-02544-t004:** Functional categories of genes differentially expressed in the hippocampus of GABA-administered mice.

Functional Categories	*p*-Value
Synapse	3.1 × 10^−7^
Zinc-finger	5.6 × 10^−5^
Postsynaptic membrane	7.3 × 10^−3^
Actin nucleation	4.5 × 10^−3^

**Table 5 nutrients-13-02544-t005:** KEGG pathway of genes differentially expressed in hippocampus of GABA-administered mice.

KEGG Pathway	*p*-Value
Long-term potentiation	2.0 × 10^−4^
Regulation of actin cytoskeleton	2.2 × 10^−4^
Alzheimer’s disease	5.9 × 10^−2^
Long-term depression	1.5 × 10^−3^

## Data Availability

The data that support the findings of this study are available from the corresponding author, Y.K., upon reasonable request.

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
