# Peer review of "Exosome-Mediated Activation of Neuronal Cells Triggered by γ-Aminobutyric Acid (GABA)"

_nutrients, 2021, doi:10.3390/nu13082544_

Round 1

Reviewer 1 Report

In regards to the manuscript by Inotsuka et al., two of the authors are from Pharma Foods International which has vested interests in gaba. They do not declare this in their conflicts of interest.

Much of the paper is not written with scientific rigor. Claims of Gaba being a relaxant, improving sleep, reducing psychological stress and other such nutraceutical claims are not well supported. In many places they jumped to conclusions—for instance they state that gabba has neuroprotective actions. The reason for this is that Gaba in the CNS declines with aging. They don't mention that tons of other neurotransmitters and proteins in the brain are declining with age. There's no reason to jump to the conclusion that mental deterioration with age is due to a loss of gaba and therefore gaba is neuroprotective. Such over statements abound and are to many to detail.

They work with a single colorectal cell line caco-2. It's not clear how this one cancerous cell line, out of the multitude of healthy cell types that constitute the gut, represents the situation in vivo.

How do they measure neurite length? Is it total neurite length? Or longest neurite length?

Please define in figure 1 what RA is. Is it retinoic acid? Please clearly define how much exosome was added to the cultures. It would enhance a manuscript if the authors showed a dose response curve.

On the microarray data: please show the heat map of which genes are going up-and-down. Just saying that genes are involved in neural regulation doesn't indicate whether they are enhancing or inhibiting neurite outgrowth.

With the in vivo data please state how long animals were treated with gaba before serum exosomes were isolated.

The most interesting thing about this report is that in vivo gaba administration produced exosomes that enhanced neurite of growth in cell cultures. That is a fascinating discovery. But to jump to conclusions in the last paragraph that microarray analysis strongly suggests that GABA administration improves memory function is again just leaping to unfounded conclusions and is typical of false claims by nutraceutical companies with vested interests.

Importantly, the authors need to state what they think is GABA's mechanism of action. Do Caco-2 cells express Gaba receptors? Do they think that activation of these receptors then changes the content of exosomes? I think they need to perform a radioactive GABA or muscimol binding assay to functionally assess whether Caco-2 cells do, or do not, functionally bind GABA. Controls could be appropriate neuronal cell lines, and negative controls COS, HEK or Hela cells as well as Caco-2 cells incubated with muscimol, baclofen and picrotoxin and saclofen (with and without GABA) should be used as a controls. Alternatively, the authors may think that Gaba itself gets transported into CACO-2 cells and is then incorporated into exosomes. In this case, they should show that Caco-2 cells take up labeled Gaba and that this gets incorporated into exosomes.

Author Response

Thank you for your valuable comments. I revised the manuscript according to the reviewer’s comments.

Reviewer #1

Comment #1:

In regards to the manuscript by Inotsuka et al., two of the authors are from Pharma Foods International which has vested interests in gaba. They do not declare this in their conflicts of interest.

Response #1:

In response to the comment, we declared the conflict of interest in the manuscript.

Comment #2:

Much of the paper is not written with scientific rigor. Claims of Gaba being a relaxant, improving sleep, reducing psychological stress and other such nutraceutical claims are not well supported. In many places they jumped to conclusions—for instance they state that gabba has neuroprotective actions. The reason for this is that Gaba in the CNS declines with aging. They don't mention that tons of other neurotransmitters and proteins in the brain are declining with age. There's no reason to jump to the conclusion that mental deterioration with age is due to a loss of gaba and therefore gaba is neuroprotective. Such over statements abound and are to many to detail.

Response #2:

Thank you for your comment. In response to the comment, we revised the manuscript.

Comment #3:

They work with a single colorectal cell line caco-2. It's not clear how this one cancerous cell line, out of the multitude of healthy cell types that constitute the gut, represents the situation in vivo.

Response #3:

Although Caco-2 cells are cancer cells, they are used in this study because they are known to exhibit differentiation functions such as digestion and absorption. This point has been added to the text.

Comment #4:

How do they measure neurite length? Is it total neurite length? Or longest neurite length?

Response #4:

We described the method to measure neurite length in the materials and methods. The total neurite length for each cell is shown in the figure.

Comment #5:

Please define in figure 1 what RA is. Is it retinoic acid? Please clearly define how much exosome was added to the cultures. It would enhance a manuscript if the authors showed a dose response curve.

Response #5:

We defined RA in Figure 1, and described the amount of exosomes added to the culture.
We had already conducted a dose response experiment in a previous experiment, and we followed it.

Comment #6:

On the microarray data: please show the heat map of which genes are going up-and-down. Just saying that genes are involved in neural regulation doesn't indicate whether they are enhancing or inhibiting neurite outgrowth.

Response #6:

In response to the comment, we added Heatmap data to the figures.

Comment #7:

With the in vivo data please state how long animals were treated with gaba before serum exosomes were isolated.

Response #7:

We have described the treatment period of GABA.

Comment #8:

The most interesting thing about this report is that in vivo gaba administration produced exosomes that enhanced neurite of growth in cell cultures. That is a fascinating discovery. But to jump to conclusions in the last paragraph that microarray analysis strongly suggests that GABA administration improves memory function is again just leaping to unfounded conclusions and is typical of false claims by nutraceutical companies with vested interests.

Response #8:

In response to the comment, we revised the manuscript.

Comment #9:

Importantly, the authors need to state what they think is GABA's mechanism of action. Do Caco-2 cells express Gaba receptors? Do they think that activation of these receptors then changes the content of exosomes? I think they need to perform a radioactive GABA or muscimol binding assay to functionally assess whether Caco-2 cells do, or do not, functionally bind GABA. Controls could be appropriate neuronal cell lines, and negative controls COS, HEK or Hela cells as well as Caco-2 cells incubated with muscimol, baclofen and picrotoxin and saclofen (with and without GABA) should be used as a controls. Alternatively, the authors may think that Gaba itself gets transported into CACO-2 cells and is then incorporated into exosomes. In this case, they should show that Caco-2 cells take up labeled Gaba and that this gets incorporated into exosomes.

Response #9:

Thank you for your important comment. In response to the comment, we revised the discussion section of the manuscript.

Reviewer 2 Report

General comments

This study by Inotsuka et al. explores the role of exosomes produced by intestinal cells in contact with GABA, on neuronal activation, both in vitro and in vivo. It follows a first study in which the authors have shown that exosomes derived from GABA-treated intestinal cells (Caco-2) serve as signal transducers that mediate gut-brain interactions. In the present study, they demonstrated that exosomes derived from GABA treated Caco-2 cells or derived from the serum of GABA treated (oral gavage) mice induced neurite growth of neuronal cells (SH-SY5Y) and increased their expression of 3 key genes (BDNF, Nestin, NEFM). The hippocampus expression of  genes implied in the memory function was also increased in these mice. This is an interesting paper, which is well written and potentially brings some very interesting results to the field but which far too elliptic in its current form to be suitable for publication. A clear recall of the state of the art on exosomes (what they are and what is known of their roles) needs to be introduced in the introduction section as well as the novelty of the present work as compared to the previous mentioned work of the authors (ref 18). The materials and methods needs to be deeply completed (see below) or at least completed with appropriate references. Some groups appearing in the results are not presented in the experimental design (RA and AICAR): what they are and their relevance should be explained. The performed statistical test is not appropriate (an ANOVA should be performed instead of a t-test regarding the number of groups). The number of wells/individuals per group is missing.  In the result section, one group appearing in the figure is never mentioned (RA) and some critical pieces of information are missing (see below). Finally, the discussion is very elliptic and some results are not discussed at all (the one regarding the mitochondria data for instance, or the miRNAs…); the authors should also be cautious when drawing some strong conclusions (see below).

Major

Introduction:

Please recall what exosomes are and what is known on their roles. Better precise the novelty of this work regarding your previous work (ref 18) as the first in vitro part of the present paper (activation of neuronal cells by exosomes secreted by Caco-2 upon GABA treatment) seems redundant with this previous paper.

Material and Methods :

Majors pieces of information are missing :

  • How the stimulation of SH-SY5Y cells by the exosomes derived from GABA-treated Caco-2 cells was performed ? How/how long were SH-SY5Y cell cultivated before their stimulation by the exosomes ? How long was the stimulation with the exosomes? What was the concentration of exosomes used ? From figure 1, one can see 2 concentrations : 500 µM and 1 mM, corresponding to the GABA concentrations used to stimulate the Caco-2 cells : do these concentrations actually correspond to the GABA concentration used to obtain the exosomes or do they correspond to the concentrations of the resulting exosomes ? If it is the GABA concentrations, how did the authors verify that the more GABA is used, the more exosomes are produced ? This needs to be clarified and precisely described. What is control condition used (named Exo-ctrl)?
  • In the same way, how were the SH-SY5Y cells stimulated by the mice serum supplemented or not by GABA? What was the exosome concentration used if controlled? The same as for the in vitro study? How long?
  • For TR-qPCR, how was the “gene expression levels normalized to the corresponding level of β-actin”? As far as I understand from figures 1 and 2, it seems that the authors made a ratio of the CT of the gene of interest and β-actin? This is not the usual way of interpreting TR-qPCR data, which is the relative quantification 2-ΔΔCt method, used to determine changes in gene expression relative to a reference sample. In this case, the reference group would be the Exo-ctrl group, and the results would be much clearer using this method, which I recommend to the authors: The ΔCt is the difference in threshold cycle between the target and reference genes:

ΔCt = Ct (target gene) – Ct (reference gene)

The ΔΔCt is then the difference in ΔCt as described in the above formula between the target samples and the reference sample: ΔΔCt= ΔCt-Ct (reference sample).

The fold change (2^(-ΔΔCt() is then compared between groups.

  • miRNA microarray assay: once again, explicite what the control is. Were all the groups used for this miRNA study (Exo-ctrl, GABA 500 µM and GABA 1 mM)?
  • Animal experiments:

The GABA dose used for mice oral administration is huge (200 mg/kg) as compared to what has been described in Humans (50-100 mg per a mean 70kg weight i.e. 1.4 mg/kg), i.e. x143! What was the rational for such a high dose? This dose was given every day for 21 days, as compared to a single dose in Human studies, what was the rational for such a long administration of the drug? Why 21 days? How many mice were there per group? Was a miRNA study performed on the serum exosomes?

  • Statistical analysis:

Can you indicate the number of individuals (wells) per group?

Considering the experimental design, at least for the in vitro studies (more than 2 groups) but also in the in vivo study in which there is a “RA” group, which is not defined so far, an ANOVA would be much more appropriate to compare all groups between each other. Please modify.

The data integration methods used to integrate the data of exosome miRNA and Exo-GABA-treated SH-SY5Y mRNA microarray has to be described here, as well as the groups used (was only one or the two doses of GABA considered?).

Results :

Exosomes derived from GABA-treated Caco-2 cells activate SH-SY5Y cells

For clarity, please always explicit the control used: i.e. as compared to Exo-ctrl (and explicit what Exo-ctrl is as it is never described, not even in the Materials and Methods section, see above). It is particularly confusing in Figure 1A in which Exo-GABA data are much less than that of RA (which is not described anywhere in the manuscript!). Please, do explain what AICAR is and why it was used as a positive control. This positive control exists only for the mitochondria characterization (number, area, activity), please explain.

  • Figure 1 : what is « Exo-ctrl » ? What is « RA » ? What is «AICAR » ? Statistical difference : compared to what group? Please indicate the number of n per group. 1E: there are problems with SD bars, please verify.
  • Figure 2: Define “BDNF, NEFM, RA”; Statistical difference: compared to what group? The significant difference between Exo-GABA 500 µM and Exo-ctrl seems really odd regarding the panels, can the authors double-check their statistics? Please indicate the number of n per group.

Molecular basis for the Exo-GABA-induced activation of SH-SY5Y

  • The four miRNAs that were changed by GABA treatment (as compared to Exo-ctrl we assume) were changed by both the concentrations of GABA (500 µM and 1 mM). Was there a dose-response? In which way these levels were changed (increase, decrease?).
  • Same questions for the mRNA expression in Exo-GABA-treated SH-SY5Y.
  • How many target genes of the four modified miRNA were there? It would be interesting to have these data in a Supplemental data table.
  • How was the comparison between these gene and the gene from the microarray analysis performed? This has also to be specified in the Material and Methods section.

Effects of exosomes derived from the serum of GABA-treated mice in SH-SY5Y cells

  • Figure 3: Define RA and AICAR, and add se-Exo-ctrl and se-Exo-GABA to be consistent with the abbreviations used in the text; statistical analysis: compared to what? Indicate the n per group.
  • Figure 4: idem; 4C: problem with SD bars, please verify;
  • 232: the augmentation is not true for Nestin gene (figure 4B), please precise.

Microarray analysis of hippocampal tissue of mice orally administered GABA

  • 243, 246: “changes”, “affected”: please be more precise and indicate what is the actual effect (increase/decrease).
  • Figure 5: L. 252-254: this piece of the legend is not clear to me, please be more explicit on the signification of the black code bar.

Discussion:

The discussion of these interesting results is very poor and disappointing since it does not consider the results in details but only in general. The authors should deeply revise their discussion including all their results (comparison with RA if pertinent, and including the mitochondria data).

The authors seem to consider that GABA is a “food component” (L. 268-270). This has to be explained and discussed.

A more precise discussion on the way the exosomes may interfere with neuronal function would be necessary, considering the role of miRNAs for instance. Do the authors know from previous works, or can they speculate, if the exosomes contain GABA or other bioactive molecules? If so, which ones?

  1. 278-280. This interesting piece of information needs to be further discussed: what are these milk exosomes? Do they contain GABA? What are their suspected way of action? Which link can be done with the present study in which GABA, and not exosomes, are ingested?
  2. 289-291. The conclusion on the role of oral GABA administration on memory function has to be moderated: here, the authors investigated in the hippocampus the effect of GABA on 3 genes, which functions should be better defined for the reader to understand the opportunity of their study (there is not a word on nestin). Gene expression does not systematically mean protein translation, so in vivo memory tests should be performed before doing such a strong conclusion. Unless doing these behavioural tests, the conclusion should be moderated.

Minor

  1. 18 : please explain what kind of cells (intestinal / neuronal) the Caco2 and SH-SY5Y cells are.
  2. 69 : Define the abbreviation « AICAR »
  3. 107-112 : define the gene abbreviation « BDNF, PGC-1α, NEFM »
  4. 156 : « three times » instead of « thrice »
  5. 265-268.: This sentence is not clear and grammatically correct and needs to be rephrased.

Author Response

Thank you for your valuable comments. I revised the manuscript according to the reviewer’s comments.

Reviewer #2

Comment #1:

Introduction

Please recall what exosomes are and what is known on their roles. Better precise the novelty of this work regarding your previous work (ref 18) as the first in vitro part of the present paper (activation of neuronal cells by exosomes secreted by Caco-2 upon GABA treatment) seems redundant with this previous paper.

Response #1:

In response to the comment, we revied the introduction.

Comment #2:

How the stimulation of SH-SY5Y cells by the exosomes derived from GABA-treated Caco-2 cells was performed ? How/how long were SH-SY5Y cell cultivated before their stimulation by the exosomes ? How long was the stimulation with the exosomes? What was the concentration of exosomes used ?

Response #2:

In response to the comment, we revised the Materials and Methods.

Comment #3

From figure 1, one can see 2 concentrations : 500 µM and 1 mM, corresponding to the GABA concentrations used to stimulate the Caco-2 cells : do these concentrations actually correspond to the GABA concentration used to obtain the exosomes or do they correspond to the concentrations of the resulting exosomes ? If it is the GABA concentrations, how did the authors verify that the more GABA is used, the more exosomes are produced ? This needs to be clarified and precisely described. What is control condition used (named Exo-ctrl)?

Response#3:

In response to the comment, we revised the manuscript (Result section).

Comment #4:

In the same way, how were the SH-SY5Y cells stimulated by the mice serum supplemented or not by GABA? What was the exosome concentration used if controlled? The same as for the in vitro study? How long?

Response #4:

In response to the comment, we revised the manuscript.

Comment # 5:

For TR-qPCR, how was the “gene expression levels normalized to the corresponding level of β-actin”? As far as I understand from figures 1 and 2, it seems that the authors made a ratio of the CT of the gene of interest and β-actin? This is not the usual way of interpreting TR-qPCR data, which is the relative quantification 2-ΔΔCt method, used to determine changes in gene expression relative to a reference sample. In this case, the reference group would be the Exo-ctrl group, and the results would be much clearer using this method, which I recommend to the authors: The ΔCt is the difference in threshold cycle between the target and reference genes:

ΔCt = Ct (target gene) – Ct (reference gene)

The ΔΔCt is then the difference in ΔCt as described in the above formula between the target samples and the reference sample: ΔΔCt= ΔCt-Ct (reference sample).

The fold change (2^(-ΔΔCt() is then compared between groups.

Response #5:

Thank you for your pertinent comments.
As the reviewer indicated, we also used the ΔΔCt method for quantification, but we made a mistake in describing the method. We have corrected the manuscript appropriately.

Comment #6:

miRNA microarray assay: once again, explicite what the control is. Were all the groups used for this miRNA study (Exo-ctrl, GABA 500 µM and GABA 1 mM)?

Response #6:

We explain the Exo-ctrl in result section. All the groups were used in this miRNA analysis.

Comment #7:

The GABA dose used for mice oral administration is huge (200 mg/kg) as compared to what has been described in Humans (50-100 mg per a mean 70kg weight i.e. 1.4 mg/kg), i.e. x143! What was the rational for such a high dose? This dose was given every day for 21 days, as compared to a single dose in Human studies, what was the rational for such a long administration of the drug? Why 21 days? How many mice were there per group? Was a miRNA study performed on the serum exosomes?

Response #7:

We determined the dosage based on previous GABA administration studies using mice. (J. Surg. Res., 236: 172 (2019), Pharmacol. Biochem. Behav., 74, 523 (2003), Jpn. J. Pharmacol., 89: 388 (2002))

In these reports, 200 mg/kg was administered systemically and about 300 mg/kg subcutaneously, and based on this, in our study, 200 mg/kg was administered orally using a sonde. In other words, we do not think that 200 mg/kg is too much.

As for the duration of administration, I think I have confused the reviewer. Acclimatization was performed for two weeks, followed by a one-week of dosing. Accordingly, we have revised the manuscript.

One group was made up of five mice, and we have revised the manuscript.

We have not performed miRNA studies using exosomes in mouse serum.

Comment #8:

Can you indicate the number of individuals (wells) per group?

Response #8:

We added n in the figure legend.

Comment #9:

Considering the experimental design, at least for the in vitro studies (more than 2 groups) but also in the in vivo study in which there is a “RA” group, which is not defined so far, an ANOVA would be much more appropriate to compare all groups between each other. Please modify.

Response #9:

In response to the comment, we redid the statistical analysis and revised the manuscript.

Comment #10:

The data integration methods used to integrate the data of exosome miRNA and Exo-GABA-treated SH-SY5Y mRNA microarray has to be described here, as well as the groups used (was only one or the two doses of GABA considered?).

Response #10:

In response to the comment, we revised the manuscript.

Comment #11:

Results:

Exosomes derived from GABA-treated Caco-2 cells activate SH-SY5Y cells

For clarity, please always explicit the control used: i.e. as compared to Exo-ctrl (and explicit what Exo-ctrl is as it is never described, not even in the Materials and Methods section, see above).

Response #11:

In response to the comment, we revised the manuscript.

Comment #12:

It is particularly confusing in Figure 1A in which Exo-GABA data are much less than that of RA (which is not described anywhere in the manuscript!). Please, do explain what AICAR is and why it was used as a positive control. This positive control exists only for the mitochondria characterization (number, area, activity), please explain.

Response #12:

In response to the comment, we revised the manuscript.

Comment #13:

Figure 1: what is «Exo-ctrl»? What is «RA»? What is «AICAR»? Statistical difference : compared to what group? Please indicate the number of n per group. 1E: there are problems with SD bars, please verify.

Response #13:

In response to the comment, we revised the manuscript.

Comment #14:

Figure 2: Define “BDNF, NEFM, RA”; Statistical difference: compared to what group? The significant difference between Exo-GABA 500 µM and Exo-ctrl seems really odd regarding the panels, can the authors double-check their statistics? Please indicate the number of n per group.

Response 14:

In response to the comment, we revised the manuscript, and replaced the Fig. 2C with new figure.

Comment #15:

Molecular basis for the Exo-GABA-induced activation of SH-SY5Y

The four miRNAs that were changed by GABA treatment (as compared to Exo-ctrl we assume) were changed by both the concentrations of GABA (500 µM and 1 mM). Was there a dose-response? In which way these levels were changed (increase, decrease?).

Response #15:

Expression of these four miRNA was commonly decreased in 500 μM and 1 mM GABA treatments. Accordingly, we revised the manuscript.

Comment #16:

Same questions for the mRNA expression in Exo-GABA-treated SH-SY5Y.

Response #16:

In response to the comment, we revised the manuscript.

Comment #17:

How many target genes of the four modified miRNA were there? It would be interesting to have these data in a Supplemental data table.

Response #17:

In response to the comment, we showed target genes in Supplemental data.

Comment #18:

How was the comparison between these gene and the gene from the microarray analysis performed? This has also to be specified in the Material and Methods section.

Response #18:

In response to the comment, we revised the manuscript.

Comment #19:

Effects of exosomes derived from the serum of GABA-treated mice in SH-SY5Y cells

Figure 3: Define RA and AICAR, and add se-Exo-ctrl and se-Exo-GABA to be consistent with the abbreviations used in the text; statistical analysis: compared to what? Indicate the n per group.

Response #19:

In response to the comment, we revised the manuscript.

Comment #20:

Figure 4: idem; 4C: problem with SD bars, please verify;

Response #20:

In response to the comment, we revied the manuscript.

Comment #21:

232: the augmentation is not true for Nestin gene (figure 4B), please precise.

Response #21:

In response to the comment, we revised the manuscript.

Comment #22:

Microarray analysis of hippocampal tissue of mice orally administered GABA

243, 246: “changes”, “affected”: please be more precise and indicate what is the actual effect (increase/decrease).

Response #22:

In response to the comment, we revised the manuscript.

Comment #23:

Figure 5: L. 252-254: this piece of the legend is not clear to me, please be more explicit on the signification of the black code bar.

Response #23:

In response to the comment, we revised the legend.

Comment #24:

Discussion:

The discussion of these interesting results is very poor and disappointing since it does not consider the results in details but only in general. The authors should deeply revise their discussion including all their results (comparison with RA if pertinent, and including the mitochondria data).

Response #24

In response to the comment, we revised the discussion.

Comment #25:

The authors seem to consider that GABA is a “food component” (L. 268-270). This has to be explained and discussed.

Response #25:

In response to the comment, we revised the discussion about GABA.

Comment #26:

A more precise discussion on the way the exosomes may interfere with neuronal function would be necessary, considering the role of miRNAs for instance.

Response #26:

In response to the comment, we revised the discussion.

Comment #27:
Do the authors know from previous works, or can they speculate, if the exosomes contain GABA or other bioactive molecules? If so, which ones?

Response #27:

We think it is necessary to verify this in the future, and we revised the discussion.

Comment #28:

278-280. This interesting piece of information needs to be further discussed: what are these milk exosomes? Do they contain GABA? What are their suspected way of action? Which link can be done with the present study in which GABA, and not exosomes, are ingested?

Response #28:

In response to the comment, we revised the discussion.

Comment #29:

289-291. The conclusion on the role of oral GABA administration on memory function has to be moderated: here, the authors investigated in the hippocampus the effect of GABA on 3 genes, which functions should be better defined for the reader to understand the opportunity of their study (there is not a word on nestin). Gene expression does not systematically mean protein translation, so in vivo memory tests should be performed before doing such a strong conclusion. Unless doing these behavioural tests, the conclusion should be moderated.

Response #29:

The discussion might confuse the reviewer.

The effect of GABA administration on brain function was inferred by GSEA analysis of gene in the hippocampus of GABA-treated mice.

The discussion has been revised to show this point.

Comment #30:

18 : please explain what kind of cells (intestinal / neuronal) the Caco2 and SH-SY5Y cells are.

Response #30

We revised the abstract.

Comment #31:

69 : Define the abbreviation « AICAR »

Response #31:

We defined AICAR.

Comment #32:
107-112 : define the gene abbreviation « BDNF, PGC-1α, NEFM »

Response #32:

We defined BDNF, PGC-1α, NEFM.

Comment #33:

156 : « three times » instead of « thrice »

Reponse #33:

We revised thrice.

Comment #34:
265-268.: This sentence is not clear and grammatically correct and needs to be rephrased.

Response #34:

In response to the comment, we revised the sentence.

Reviewer 3 Report

Dear authors,

The effort to submit the manuscript titled, Exosome-mediated activation of neuronal cells triggered by γ-aminobutyric acid (GABA) is appreciated. Overall, the manuscript reads well, is an interesting, and up-to-date study about GABA. However, the current form needs some minor revision and clarity. The manuscript could be strengthened by attention to these areas:

  1. It would be interesting to include and present the mechanism of action of GABA in a form of a figure or diagram.
  2. In the introduction, the relation between the immune system and GABA has not been mentioned although GABA has been reported to be an active immunomodulatory molecule.
  3. In addition to the neurological and psychiatric disorders linked to a decrease amount of GABA, it has also been associated with several health-promoting properties that were not mentioned in the manuscript such as anti-diabetes, anti-cancer, antioxidant, anti-inflammation, anti-microbial, and anti-allergy…etc.
  4. The authors mentioned that the function and levels of GABA were found to decrease with aging. Are there any other factors that affect low levels of GABA?
  5. Line 41, the authors mentioned the recent emphasis on GABA-rich functional foods due to its reported bioactivities. Could you give some examples?
  6. Does the way of administration of GABA affect its function? For example, oral administration through supplements or through GABA-rich foods?
  7. What is the relation between GABA and microRNA?
  8. The authors have not referred to GABA receptors or the role of G proteins in the activation of GABA receptors in the manuscript.
  9. Cryo-Electron Microscopy (cryo-EM) studies could be considered to further understand the molecular mechanisms and structure by which GABA functions.
  10. Although nearly half of the cited references are from the last five years, the authors cite several very old studies (published more than 20 years ago). Are these really necessary?

Author Response

Thank you for your valuable comments. I revised the manuscript according to the reviewer’s comments.

The effort to submit the manuscript titled, Exosome-mediated activation of neuronal cells triggered by γ-aminobutyric acid (GABA) is appreciated. Overall, the manuscript reads well, is aninteresting, and up-to-date study about GABA. However, the current form needs some minor revision and clarity. The manuscript could be strengthened by attention to these areas:

Comment #1:

It would be interesting to include and present the mechanism of action of GABA in a form of a figure or diagram.

Response #1:

In response to the comment, we added schematic diagram of GABA function.

Comment #2:

In the introduction, the relation between the immune system and GABA has not been mentioned although GABA has been reported to be an active immunomodulatory molecule.

Response #2

In response to the comment, we revised the introduction.

Comment #3:

In addition to the neurological and psychiatric disorders linked to a decrease amount of GABA, it has also been associated with several health-promoting properties that were not mentioned in the manuscript such as anti-diabetes, anti-cancer, antioxidant, anti-inflammation, anti-microbial, and anti-allergy…etc.

Response #3:

In response to the comment, we revised the introduction.

Comment #4:

The authors mentioned that the function and levels of GABA were found to decrease with aging. Are there any other factors that affect low levels of GABA?

Response #4:

In response to the comment, we revised the introduction.

Comment #5:

Line 41, the authors mentioned the recent emphasis on GABA-rich functional foods due to its reported bioactivities. Could you give some examples?

Response #5:

In response to the comment, we added example of GABA-rich functional foods.

Comment #6:

Does the way of administration of GABA affect its function? For example, oral administration through supplements or through GABA-rich foods?

Response #6:

I think it is necessary to verify whether the way GABA is administered affects the function or not. We would like to verify this in the future.

Comment #7:

What is the relation between GABA and microRNA?

Response #7:

The detailed mechanism is not known, but as shown in Fig. 7, GABA treatment changes the miRNAs in the exosomes produced by intestinal cells to contain more miRNAs that promote neuronal activation.

Comment #8:

The authors have not referred to GABA receptors or the role of G proteins in the activation of GABA receptors in the manuscript.

Response #8:

In response to the comment, we revised the introduction.

Comment #9:

Cryo-Electron Microscopy (cryo-EM) studies could be considered to further understand the molecular mechanisms and structure by which GABA functions.

Response #9:

Thank you for your comment. We would like to clarify the molecular mechanisms of GABA functions in the future study.

Comment #10:

Although nearly half of the cited references are from the last five years, the authors cite several very old studies (published more than 20 years ago). Are these really necessary?

Response #10.

In response to the comment, we replaced old references with new ones.